# Transversus Abdominis Ultrasound Thickness during Popular Trunk–Pilates Exercises in Young and Middle-Aged Women

**DOI:** 10.3390/jfmk8030110

**Published:** 2023-08-04

**Authors:** Ioannis Tsartsapakis, Maria Gerou, Aglaia Zafeiroudi, Eleftherios Kellis

**Affiliations:** 1Laboratory of Neuromechanics, Department of Physical Education and Sport Sciences at Serres, Aristotle University of Thessaloniki, 62100 Serres, Greece; ioantsar@phed-sr.auth.gr (I.T.);; 2Department Physical Education & Sport Science, University of Thessaly, 42100 Trikala, Greece

**Keywords:** transversus abdominis, abdominal draw-in maneuver, ultrasound imaging, TrA thickness, kinetic and functional stabilization

## Abstract

The transversus abdominis (TrA) is a core muscle that contributes to functional mobility and lumbar stability. This study aimed to compare the changes in TrA thickness during different Pilates exercises, and to identify the exercise that elicited the greatest TrA activation. Forty-four healthy women were divided into two groups: young (25–35 years old) and middle-aged (36–55 years old). TrA thickness was assessed by ultrasound while the participants performed five Pilates exercises: basic position, hundred, hip roll, side plank, and dead bug. A repeated measures analysis of variance revealed that the dead bug exercise induced a significantly higher increase in TrA thickness (relative to rest) than the other exercises (*p* < 0.05). The young group also showed a significantly higher overall TrA thickness than the middle-aged group (*p* < 0.05). The findings suggest that the dead bug exercise is the most effective for enhancing TrA activation among the Pilates exercises tested. The basic position and the hundred exercises can be used as warm-up exercises before performing more challenging exercises such as the hip roll, the side plank, and the dead bug. The sequence of exercises can be similar for both young and middle-aged women.

## 1. Introduction

Core stability and trunk control are essential for maintaining postural alignment, preventing injuries, and enhancing performance in various physical activities [1]. The core musculature consists of the muscles of the pelvic floor, the abdominal and back muscles, the diaphragm, and the transversus abdominis (TrA) [2]. The TrA is considered to be the primary stabilizer of the lumbar spine and the pelvis, as it is activated prior to any movement of the limbs [3]. The TrA also reduces the laxity of the sacroiliac joint and contributes to flattening the abdomen [4]. However, the TrA is often underutilized or dysfunctional in many individuals, especially those with low back pain or core instability [4]. Therefore, training the TrA and improving its activation during various movements is important for enhancing core function and reducing pain [5].

One of the methods to assess the function of the TrA is ultrasound (US) imaging, which can measure the thickness of the muscle during rest and contraction [6]. US imaging has been shown to be a reliable and valid technique to evaluate TrA activation, as it reflects the changes in muscle fiber length and pennation angle [7]. US imaging also has some advantages over other methods of measuring TrA function, such as electromyography (EMG) or pressure biofeedback, as it is non-invasive, easy to use, and provides real-time feedback [8]. However, US imaging also has some limitations, such as being dependent on operator skill, requiring standardized positioning and calibration, and being influenced by factors such as hydration status and body fat [9].

Pilates is a popular exercise method developed by Joseph Pilates with its own philosophy and basic principles. The roots of Pilates come from ancient Greek and Eastern philosophy about mind body and spirit connection [10,11]. Today Pilates is divided into two types, the classic and the modern or contemporary, with many differences between them [11]. Classical Pilates exercises focus on posterior pelvic tilt, and they are taught as Joseph Pilates originally invented them without any modification and in the same order every time [11]. In contrast, contemporary Pilates incorporated new exercises or modifications of the existing ones whilst they are performed with a neutral pelvis position. There are suggestions that contemporary Pilates consider latest scientific developments to make the method more appropriate, functional, and safer for the participants [11]. In fact, in contemporary Pilates, any movement and exercise can be incorporated according to the basic principles and philosophy of the method.

The Pilates method constitutes a fundamental component in the movement, health and fitness, and rehabilitation industry. It focuses on training the core musculature, especially the TrA [12]. Pilates exercises are based on six principles: centering, concentration, control, precision, breath, and flow [13]. Exercises that are used in Pilates programs involve performing movements with a neutral spine alignment and a co-contraction of the deep abdominal and pelvic floor muscles [14]. Some examples of common exercises are the supine hook lying abdominal drawing-in maneuver (ADIM) [15], the hundred A and B exercises, which involve holding a static position with arm and leg movements while breathing deeply [16], and the leg raise and roll up exercises, which challenge the trunk stability and mobility while moving the legs or the upper body [17]. The ADIM is a slow and gentle abdominal hollowing [18] maneuver where individuals gently pull the umbilicus towards the spine, hold the contraction, and breathe normally [19,20,21] for up to 10 s. Originally, the ADIM is performed after the participants assume a supine position with their hips at 40–60° and their knees are flexed between 90° and 100° and their arms are placed along their torso [18]. Since then, the ADIM is performed simultaneously with other exercises to increase TrA contraction [19,20,21].

Due to its deep anatomical location, non-invasive examination of TrA activation is difficult. For this reason, ultrasound (US) thickness measurements have been used as an indirect index of muscle size at rest as well as an index of muscle recruitment during exercise [22,23]. Core stability exercises have been shown to increase TrA thickness in healthy women and improve low back pain in women with chronic pain [24]. However, there is limited evidence on which Pilates exercises produce the greatest activation of the TrA and whether this activation differs according to the age of the participants. Previous studies have reported conflicting results on which exercises elicit higher TrA recruitment, such as upper or lower limb lifts from the quadruped position [25], prone plank [26], bridge exercises with abdominal bracing [27,28], or planks with opposite arm–leg elevation [29]. Moreover, some studies have suggested that there may be age-related differences in TrA activation [30,31]. A study by Kellis et al. [6] showed that young adults had the greatest relative transversus abdominis thickness during contraction compared to middle-aged adults, adolescents, and children. The aim of this study was to compare TrA thickness during five common exercises (basic position, hundred, hip roll (or bridge), side plank, and dead bug) that are used in Pilates programs and to examine whether there are differences in TrA thickness between younger and middle-aged (but more experienced) participants. It was hypothesized that (1) different Pilates exercises would result in different levels of TrA activation and (2) young participants would have higher TrA activation than middle-aged participants.

## 2. Materials and Methods

### 2.1. Participants

A total of 44 adult women mean age 33.2 ± 8.7 (25 to 55 years of age) participated in this study. The participants were divided into two groups: the young group (N = 29, 27.7 ± 2.4 years) and the middle-aged group (N = 15, 44.0 ± 5.7 years). The participants did not report any problems (pain or chronic low back pain) in the area of the core and the trunk. They also had at least three years of experience with Pilates programs.

The study was conducted in accordance with the ethical guidelines of Aristotle University of Thessaloniki, and all procedures followed the most recent version of the Declaration of Helsinki. All participants were informed about the purpose of the research and provided their written consent to participate.

### 2.2. Instrumentation

All TrA measurements were collected with an Aloka Prosound SSD-3500SV US system (Aloka Inc, Tokyo, Japan). The transducer head had a length of 6 cm, and its frequency was 13 MHZ.

### 2.3. Exercise Protocol

The duration of the entire procedure was 50 min for each participant to avoid fatigue in the trunk area. The participants first received relevant information on how to perform each of the exercises and then performed several trials of each exercise to gain familiarization. Following familiarization, the main measurements were collected.

First, the participants assumed the relaxed position (Figure 1A) by lying on the mat with hands on their sides and their knees flexed 90°. TrA muscle thickness was measured at rest in this position. Then, a representative set of five classical Pilates mat exercises were performed. The examiner first asked the participants to assume one of the five exercise postures.

In particular, the following instructions were given to the participants:

(1) Relax and basic Pilates position (Figure 1A,B) “lie on your mat with your legs bent, feet flat on the ground, chin in toward chest, pelvis in a neutral position. Feel the spine on your mat keeping the natural curve of the lumbar spine. Shoulders are relaxed away from the ears; arms are relaxed with the palms touching the ground”. In the basic Pilates position, participants were asked to perform the ADIM for voluntary TrA activation three times.

(2) Hundred (Figure 1C), “lie on your mat with your legs bent, feet flat on the ground and lengthen your spine, pelvis in neutral position. Inhale activate your abs. Exhale lift your head and shoulders off the mat with your eyes between your legs. Arms are off the mat next to the sides of the body, palms facing down, fingers outstretched, and shoulders fixed away from the ears. Arms pump vigorously, lifting up and down no higher than the hips. While pumping arms, inhale and exhale 5 times to complete one repetition. The exercise is performed 10 times with 5 inhales and 5 exhales equaling 100”.

(3) Hip roll (Figure 1D), “lie on your back with your knees bent and your legs hip-width apart. Knees are bent at about 90 degrees. Arms are straight and palms down on the mat by your sides. Pelvis should be in a neutral position. This means that your pelvis and bum is neither tucked under (so that your back is flat) nor is it duck-like and sticking out. Take a deep, full inhale and create space in your spine by imagining it lengthening. Then start to exhale slowly through an open mouth. Press into your feet and start to peel your spine up into the bridge position starting from your tailbone. Only bridge up to the point where your shoulder blades are resting on the mat. Pause at the top of your bridge and take an inhale here. Exhale and slowly start to bridge back down. To initiate this part of the movement, first allow your chest to soften. Then allow the rest of the spine to trickle down segmentally to the mat vertebra by vertebra. Your bridge is complete when your body is back resting on the mat with your pelvis in neutral position. Three repetitions will be performed”.

(4) Side plank (Figure 1E), “lie on your left side with one leg straight and the other with the knee on the mat. Place your left elbow on the mat. The elbow of your left arm is directly under your shoulder. Spine is in neutral position. Ensure your head is directly in line with your spine. Stretch the right arm toward the ceiling. Engage your abdominal muscles, drawing your navel toward your spine. Your torso is straight in line with no sagging or bending. Hold the position for 3 breaths. Repeat on the other side. Three repetitions will be performed”.

(5) Dead bug (Figure 1F), “lie on the mat with your arms extended straight over your chest so they form a perpendicular angle with your torso. Bend your hips and knees 90-degrees, lifting your feet from the ground. Your torso and thighs should form a right angle, as should your thighs and shins (table top). Engage your core, maintaining contact between your lower back and the mat. Your spine maintains in neutral position throughout the exercise. Three repetitions will be performed”. All instructions were given by the same instructor who is a certified Pilates teacher.

Except the resting position, every participant was asked to perform the ADIM while they executed each exercise for a period of ten seconds.

### 2.4. Muscle Thickness Measurements

The US head was placed 2.5 cm above the iliac crest and along the axillary line [32]. The position of the head was standardized by placing the anterior origin of the fascia 2 cm to the left of the middle of the US image when relaxed [33]. US images were used for analysis during the rest condition and while participants performed the ADIM during each exercise. Using the manufacturer’s US device software (v2.0, Aloka Inc, Tokyo, Japan), the US image was frozen approximately between 4 and 6 s of the 10 s rest condition or ADIM contraction. TrA was measured on the US device screen as the distance from the superior to inferior fascia of the muscle 2.5 cm from the anterior origin of the fascia (Figure 2). Three repetitions were analyzed, and an average muscle thickness measurement was calculated. In addition to absolute thickness (measured in mm), TrA relative thickness ratio during each exercise was expressed as a percentage of resting thickness, (TrA active − TrA rest)/TrA rest × 100, [34] for each exercise (Figure 3).

In a previous study, the inter-examiner and intra-examiner reliability of TrA muscle thickness at rest and contraction was examined, and the reliability was high with intraclass correlation coefficients ranging from 0.86 to 0.97 at rest, from 0.89 to 0.97 during ADIM, and from 0.77 to 0.98 for the relative thickness values [6].

### 2.5. Statistical Analysis

All statistical analyses were conducted using IBM SPSS Statistics ver. 26.0 (IBM Co., Armonk, NY, USA). Normal distribution of the collected data was verified by using the Kolmogorov–Smirnov test. An independent sample *t*-test was performed to evaluate whether there was a statistically significant difference in anthropometric characteristics and years of Pilates training between the two groups. A two-way repeated measures ANOVA was applied to examine the effects of exercise (basic position, hundred, hip roll, side plank, and dead bug) and group (young vs. middle-aged) on absolute and relative TrA thickness. The Greenhouse–Geisser correction was used when the assumption of sphericity was violated. The degrees of freedom (df), mean square error (MSE), F or t values, and effect sizes (η^2^ or Cohen’s d) were reported for each test. Post hoc comparisons were performed using the Bonferroni correction. The level of significance was set at *p* < 0.05.

## 3. Results

### 3.1. Descriptives

The anthropometric characteristics and years of Pilates training of the entire sample and each group are shown in Table 1. No significant differences were found in anthropometric characteristics between the two groups (*p* > 0.05).

### 3.2. Absolute Thickness

The mean absolute thickness descriptive values for each exercise and group are presented in Table 2. A two-way repeated measures ANOVA revealed a significant main effect of exercise on TrA muscle absolute thickness (*F*_(4, 168)_ = 230.37, *p* < 0.001, η*_p_*^2^ = 0.84), indicating that TrA thickness varied across different exercises. Post hoc Bonferroni analysis indicated that the dead bug exercise had a significantly greater mean TrA absolute thickness score than the side plank, hip roll, hundred, and the basic position exercises, respectively, (*p* < 0.001). Also, the side plank exercise had a significantly greater mean TrA thickness score than the hip roll, hundred, and the basic position exercises (*p* < 0.001). Likewise, the hip roll exercise had a significantly greater mean TrA thickness score than hundred and the basic position exercises (*p* < 0.001). Lastly, the hundred exercise had a significantly greater mean TrA thickness score than the basic position exercise (*p* < 0.001).

A significant main effect of group on TrA muscle absolute thickness was also found (*F*_(1, 42) =_ 16.14, *p* < 0.001, η*_p_*^2^ = *0*.27), with the young group had a significantly greater mean TrA thickness score in all exercises than the middle-aged group (*p* < 0.001).

No significant interaction effect between exercise and group on TrA muscle absolute thickness was found (*F*_(4, 168) =_ 0.75, *p* = 0.55), indicating that the effect of exercise on TrA thickness was similar for both groups.

### 3.3. Relative Thickness

The mean relative thickness descriptive values for each exercise and group are presented in Figure 3. A two-way repeated measures ANOVA revealed a significant main effect of exercise on TrA muscle relative thickness (*F*_(4, 168) =_ 142.84, *p* < 0.001, η*_p_*^2^ = 0.87), indicating that TrA relative thickness varied across different exercises. Post hoc Bonferroni analysis indicated that the dead bug exercise had a significantly greater mean TrA relative thickness score than the side plank, hip roll, hundred and the basic position exercises (*p* < 0.001). Also, the side plank exercise had a significantly greater mean TrA relative thickness score than the hip roll, hundred, and the basic position exercises (*p* < 0.001). Likewise, the hip roll exercise had a significantly greater mean TrA relative thickness score than hundred and the basic position exercises (*p* < 0.001). Lastly, the hundred exercise had a significantly greater mean TrA relative thickness score than the basic position exercise (*p* < 0.001).

A significant main effect of group on TrA muscle relative thickness was also found (*F*_(1, 42) =_ 7.01, *p* < 0.05, η*_p_*^2^ = 0.14), with the young group having a significantly greater mean TrA thickness score in all exercises than the middle-aged group (*p =* 0.011*)*.

No significant interaction effect between exercise and group on TrA muscle relative thickness was found (*F*_(4, 168)_ = 0.75, *p* = 0.55), indicating that the effect of exercise on TrA relative thickness was similar for both groups.

## 4. Discussion

The purpose of the present study was to examine TrA thickness in a series of exercises that are an integral part of the Pilates exercise programs. Further, it was examined whether a difference in TrA thickness during each exercise exists between the younger and middle-aged participants. The main findings of this study: (a) the dead bug exercise resulted in the highest TrA activation, followed by the side plank, hip roll, hundred and basic position exercises; and (b) the young group had higher TrA activation than the middle-aged group in all exercises.

### 4.1. Differences between Exercises

Our results showed that there was a ranking of the exercises in terms of TrA activation, which was the same for both groups. The dead bug exercise elicited the highest TrA activation, followed by the side plank, hip roll, hundred, and basic position exercises. This ranking suggests that exercises that require the co-activation of other muscle groups besides the core (dead bug) or exercises that involve lifting the trunk off the ground (side plank and hip roll) induce greater TrA activation than exercises that keep the trunk in contact with the ground and do not require co-activation of large muscle groups (basic position and hundred). This finding is consistent with previous studies that have reported higher TrA activation during exercises that challenge the trunk stability and mobility, such as planks [26], hip roll [27], or limb lifts [25], compared to exercises that involve less trunk movement or load, such as curl-ups [26] or hook lying [27].

One possible explanation for this finding is that the TrA plays a key role in stabilizing the lumbar spine and pelvis during movements or perturbations that impose high loads or forces on these regions [35]. The TrA can stabilize the spine by increasing the intra-abdominal pressure [36] or by resisting the rotational and translational forces during trunk movements [37]. Therefore, exercises that involve lifting or moving the trunk or limbs may require higher TrA activation to maintain spinal stability and alignment than exercises that keep the trunk in a neutral position. Another possible explanation is that Pilates exercises are based on six principles: centering, concentration, control, precision, breath, and flow [11]. These principles emphasize performing movements with a neutral spine alignment and a co-contraction of the deep abdominal and pelvic floor muscles [14]. Therefore, Pilates exercises may facilitate higher TrA activation by enhancing the awareness and coordination of these muscles during various movements.

Our results are partly in agreement with previous studies that have examined which core stability exercises produce the greatest TrA activation [25,26,27,29]. However, some studies have reported different results, such as higher TrA activation during prone plank [26], hip roll with abdominal bracing [27], or plank with opposite arm-leg elevation [29] than during other exercises. These discrepancies may be due to several factors, such as the use of different methods of measuring TrA activation (US vs. EMG), the use of different populations (healthy vs. low back pain), or the use of different types of exercises (Pilates vs. non-Pilates). Therefore, more studies are needed to compare TrA activation during various Pilates and non-Pilates exercises using both US and EMG methods in different populations.

### 4.2. Differences between Groups

Our results showed that there was no difference in TrA thickness at rest between the two groups. This finding is in line with two previous studies that have shown that age-related atrophy is less pronounced in deep trunk muscles such as the transversus abdominis and lumbar multifidus [38,39]. However, our results also show that young adults had significantly greater relative average TrA thickness than the middle-aged group in all exercises. This finding is consistent with previous research [6,30,31] as well as with a systematic review that has shown that younger individuals can recruit TrA higher than older individuals [40]. Our findings may be explained by several factors, such as differences in neuromuscular control, muscle fiber type composition, or hormonal status between younger and older individuals [40]. Also, it may be possible that other factors, such as genetic predisposition, lifestyle habits, or injury history, may have influenced TrA activation. Therefore, more studies are needed to examine the effects of Pilates experience on TrA activation using larger samples and longitudinal designs.

### 4.3. Implications and Applications

The results of this study have some implications and applications for clinical practice and future research. First, our results suggest that exercises that are part of Pilates programs are effective in activating the TrA during various movements. This is important because previous studies have shown that such exercises can improve core stability and function [12], reduce low back pain [41], and enhance quality of life [10] in different populations. Therefore, Pilates programs including such exercises may be beneficial for individuals who want to improve their core health and performance. Second, our results suggest that different Pilates exercises result in different levels of TrA activation. This is important because it can be used in the design and progression of Pilates programs for different purposes and populations. For example, exercises that elicit lower TrA activation (basic position and hundred) may be suitable for beginners or individuals with low back pain who need to learn how to activate their core muscles properly. Exercises that elicit higher TrA activation (dead bug, side plank, and hip roll) may be suitable for advanced or healthy individuals who want to challenge their core stability and mobility. Third, our results suggest that younger individuals can activate their core muscles higher than older individuals regardless of their Pilates experience. This is important because it indicates that age may be a more influential factor than experience on core muscle function. Therefore, more attention should be paid to training older individuals who may have reduced core muscle function due to aging.

### 4.4. Limitations

The present study has several limitations. First, the study was conducted with recreationally active females with no musculoskeletal injuries. This limits the generalizability of the results to other ages, genders, and populations who may have different levels of core muscle function or pathology. Second, the study examined only one muscle (TrA) using one method (US). This limits the comprehensiveness of the assessment of core muscle function as there are other muscles involved in core stability (such as internal oblique and multifidus) and other methods to measure muscle activity (such as EMG). Third, the study examined only static exercises performed on a stable surface. This limits the ecological validity of the assessment of core muscle function as there are many dynamic movements performed on unstable surfaces in daily life or sports activities. Further, US muscle thickness measurements may be influenced by body fluid shifts [42]. In the present study, the participants were instructed to maintain standard hydration whilst exercises were performed from a supine position, which most likely involves minimal body fluid shifts [42], and, hence, it is likely that the effect of body fluids on muscle thickness measurements was minimal.

## 5. Conclusions

We examined TrA thickness during five common exercises that are part of Pilates programs performed by young and middle-aged women with Pilates experience. Of these exercises, those that elicited higher TrA activation were dead bug exercise followed by side plank exercise followed by hip roll exercise followed by hundred exercise followed by basic position exercise. Younger women had higher TrA activation than middle-aged women. These findings could be used to adjust exercise progression during a core stabilization exercise program based on age and purpose. Future studies should examine more core muscles using other methods during dynamic movements performed on stable and unstable surfaces in different populations.

## Figures and Tables

**Figure 1 jfmk-08-00110-f001:**
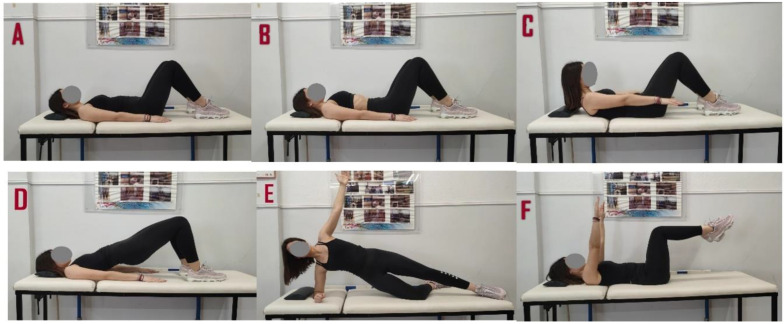
Illustration of the examined exercises: (**A**) relax position, (**B**) basic position, (**C**) hundred, (**D**) hip roll, (**E**) side plank, and (**F**) dead bug.

**Figure 2 jfmk-08-00110-f002:**
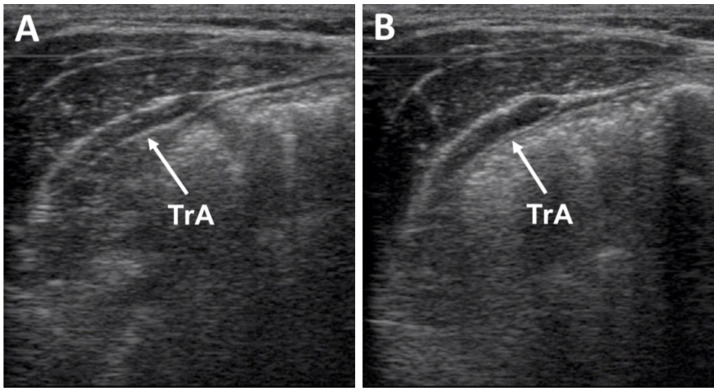
Ultrasound screenshot of TrA measurement. (**A**) TrA at rest prior to the application of ADIM. (**B**) TrA during a contraction after the application of ADIM.

**Figure 3 jfmk-08-00110-f003:**
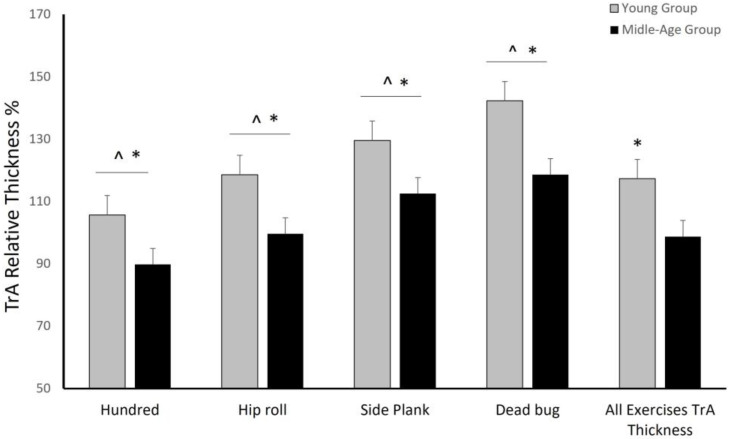
Mean (±SE) TrA relative thickness in each exercise for both groups, expressed as a percentage of the thickness at rest. * Young group had a significantly higher relative mean TrA thickness in all exercises than the middle-aged group (*p* =< 0.001); ^ each exercise differed compared to other exercises, (*p* < 0.01).

**Table 1 jfmk-08-00110-t001:** Mean (±SD) anthropometric characteristics of each group (BMI = body mass index).

	Total Sample	Young	Middle Age
N	44	29	15
Age (years)	33.2 ± 8.6	27.7 ± 2.4	44.0 ± 5.7
Height (cm)	166.1 ± 6.3	166.6 ± 7.0	165.1 ± 4.9
Mass (kg)	59.0 ± 5.6	58.3 ± 5.5	60.3 ± 5.7
BMI	214 ± 1.9	21.0 ± 1.7	22.1 ± 2.0

**Table 2 jfmk-08-00110-t002:** Mean (± SD) group absolute TrA thickness values for each exercise in the entire sample, the young participants group (Young) and the middle-aged participants group (Middle-Aged). N = number of participants.

	Total Sample	Young	Middle-Aged
N	44	29	15
Basic position	6.68 ± 0.70 ^	6.92 ± 0.70 *	6.23 ± 0.47
Hundred	7.25 ± 0.73 ^	7.47 ± 0.72 *	6.83 ± 0.55
Hip roll	7.64 ± 0.68 ^	7.89 ± 0.64 *	7.17 ± 0.50
Side plank	8.07 ± 0.64 ^	8.29 ± 0.62 *	7.64 ± 0.46
Dead bug	8.41 ± 0.68 ^	8.70 ± 0.60 *	7.86 ± 0.48
All exercises TrA thickness	7.61 ± 0.68	7.85 ± 0.70 *	7.15 ± 0.65

* Young group had a significantly higher absolute mean TrA thickness in all exercises than the middle-aged group (*p* < 0.001), ^ significantly different compared to other exercises, (*p* < 0.01).

## Data Availability

The data presented in this study are available on request from the corresponding author.

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
