# Peer review of "Transversus Abdominis Ultrasound Thickness during Popular Trunk–Pilates Exercises in Young and Middle-Aged Women"

_jfmk, 2023, doi:10.3390/jfmk8030110_

Round 1

Reviewer 1 Report

The present study investigates the transversus abdominis activation during five different Pilates exercises in young and experienced middle-aged women. 

The study is well conducted and described, although it presents some limits.  By the way, it needs only minor revisions:

1. in the introduction, Pilates is extensively described. However, due to its evolution in recent years, it is important to define if authors refer to classic Pilates in the introduction and maybe furnish some examples about different kinds of Pilates.

2. Authors refer to the ADIM technique. It should be described better in the main text or the supplementary material and refer to it.

3. the third hypothesis is a potential application and not a hypothesis that can be demonstrated. Please, move these sentence in the discussion.

4. Please, check that TrA is not written TRA throughout the text.

5. Why Basic position (ADIM)? it seems in the introduction that ADIM is performed just on the basic position. Then this is clarified in the methods section, but the information seems not coherent. I would suggest writing the name of all the positions and specifying that ADIM was performed for each of them, while in the basic position, the assessment was performed with and without ADIM.

6.  From figure 3 it seems that only the last exercise is different between groups.

7. The authors illustrated some of the limitations of the US technique without discussing them. First, why do not assess hydration using bioelectrical impedance vector analysis or urine specificity gravity methods? Alternatively, why do not control for hydration, asking participants to reach a state of euhydration in the days before the assessment? this limit should be discussed. Also, the technique is dependent on operator skills. The study could be performed with two or more operators to verify the consensus. Please, also discuss this aspect.

Author Response

Thank you for your comments. We have revised the paper accodingly.

Comment 1. in the introduction, Pilates is extensively described. However, due to its evolution in recent years, it is important to define if authors refer to classic Pilates in the introduction and maybe furnish some examples about different kinds of Pilates.

Author response: We accept the comment and we included a paragraph explaining the Pilates method in more detail for the reader (Lines 45-56)

Comment 2. Authors refer to the ADIM technique. It should be described better in the main text or the supplementary material and refer to it.

Author response: We agree and added an explanation of the ADIM (Lines 66 – 72).

Comment 3. Τhe third hypothesis is a potential application and not a hypothesis that can be demonstrated. Please, move these sentence in the discussion.

Author response:   It moved to the applications paragraph.

Comment 4. Please, check that TrA is not written TRA throughout the text.

Author response: It was checked throughout.

Comment 5. Why Basic position (ADIM)? it seems in the introduction that ADIM is performed just on the basic position. Then this is clarified in the methods section, but the information seems not coherent. I would suggest writing the name of all the positions and specifying that ADIM was performed for each of them, while in the basic position, the assessment was performed with and without ADIM.

Author response: The ADIM is now described and clarified (Lines 66-72). It is also clarified that the ADIM is performed simultaneously with other exercise movements, not just in the basic position (Lines 164-165).

Comment 6. From figure 3 it seems that only the last exercise is different between groups.

Author response:  Asterisks were added in each exercise (Figure 3).

Comment 7. A) The authors illustrated some of the limitations of the US technique without discussing them. First, why do not assess hydration using bioelectrical impedance vector analysis or urine specificity gravity methods? Alternatively, why do not control for hydration, asking participants to reach a state of euhydration in the days before the assessment? this limit should be discussed. B) Also, the technique is dependent on operator skills. The study could be performed with two or more operators to verify the consensus. Please, also discuss this aspect.

Author response:

  1. Indeed body fluid shifts can have an influence on US B-mode image measurements whilst dehydration mainly influences body fat thickness measurements using US which was not performed in this study. A new sentence was added in the Limitations section, acknowledging this (Lines 342-347).
  2. We agree. We have already checked and published the reliability of TrA measurements using US experimental set-up of our laboratory with high intra- and inter-examiner reliability. This is now mentioned in the text (Lines 180-183)

Reviewer 2 Report

General:

This a solid study that answers a valuable question. The paper is well-written and used good methods. Additionally, the study is a good fit with the journal.

Therefore, I have mostly minor comments for the authors to consider.

The biggest issues involve when the measurements were taken, and measurement reliability (see below).

Title:

The title is good with no changes needed. However, I do wonder if the exercises chosen are truly ‘Pilaties’ exercises, or simply exercises used in Pilaties? Perhaps consider ‘popular abdominal’ or ‘popular trunk’ exercises.

Abstract:

Great job shaping the topic and question. However, I recommend a few works explaining how acute changes in thickness can be used as a proxy for muscle activation (like EMG).

Introduction:

As mentioned in the title, I am not sure if some/all of the chosen exercises are specific to ‘Pilates’, or are simple common core movements. Some like the ‘hundred’ clearly are, but other like the side plank, and dead bug are not (at least to my knowledge). A simple statement from the authors to this regard might be valuable/clarifying.

Methods:

Line 74: I would report the mean and SD first, then the range in brackets.

Check for consistency in reporting numbers and symbols (spaces between them, or not)

Figure 1 needs more explanation. You mention the ultrasound hardware, then refer to the image, which is of an actual muscle thickness image. Either move the image to later in the manuscript where the image evaluation takes place, or include far more detail regarding figure 1 in the same paragraph as it is in currently. I personally recommend moving that present small paragraph to the beginning of the ‘Muscle thickness measurements’ subsections (currently lines 109-117)

The ’hundred’ exercise needed more explanation. Currently, the authors state, “In the Hundred exercise, individuals were asked to lift their heads and shoulders towards their hips and then performed the ADIM”. Why is it called ‘the hundred’? It is held for 100 seconds, or does the person do 100 reps etc? (CORRECTION: I do see these instructions in the appendix, however, I would now recommend adding them to the body of the text, and removing the appendix..)

Their also needs to be more information regarding WHEN the images are collected. For example, how many seconds after starting the side place is the image taken? How many images are taken? Are they averaged? Do we know the researchers' reliability (this is esp. important given the authors state in the intro that US imaging is dependent on practitioner skill)?

The statistical analysis is good, though I also would like to see post hoc (pairwise) effect sizes

Results:

The authors can reduce the number of significant digits (esp for F scores) to improve readability.

Figure 3 can be greatly improved by providing brackets (or another illustration) to highlight significant differences (with ESs) between exercises. This would make the figure an excellent way of summarizing the entire results section!

Discussion:

I have no issues with the discussion. Nice work!

Author Response

Thank you very much for you comments which helped to improve the manuscript

Title:

Comment 1. The title is good with no changes needed. However, I do wonder if the exercises chosen are truly ‘Pilaties’ exercises, or simply exercises used in Pilaties? Perhaps consider ‘popular abdominal’ or ‘popular trunk’ exercises.

Author response: Thank you for pointing this out. We used this title because many people refer to these exercises as such, so the title may “target” these audience. Nevertheless, we agree with this comment and we have made two actions: 1. We changed “Pilates Exercises” to “Popular Trunk / Pilates exercises” in the title and 2. We have made several modifications within the article by changing the expression “Pilates Exercises” to “Exercises that are used in Pilates Programs” or similar expressions.

Abstract:

Comment 2. Great job shaping the topic and question. However, I recommend a few works explaining how acute changes in thickness can be used as a proxy for muscle activation (like EMG).

Author response: We have modified the text to introduce the reader to the use of US for measuring TrA thickness (as an indirect measure of muscle activation) (Lines 73-75).

Introduction:

Comment 3. As mentioned in the title, I am not sure if some/all of the chosen exercises are specific to ‘Pilates’, or are simple common core movements. Some like the ‘hundred’ clearly are, but other like the side plank, and dead bug are not (at least to my knowledge). A simple statement from the authors to this regard might be valuable/clarifying.

Author response: We already addressed this comment before. In short, we agree and we have made several modifications referring to "Exercises that are used in Pilates" and not "Pilates exercises".

Methods:

Comment 4. Line 74: I would report the mean and SD first, then the range in brackets.

Author response: Done (Line 97)

Comment 5. Check for consistency in reporting numbers and symbols (spaces between them, or not).

Author response: We have scanned the manuscript and revise throughout.

Comment 6. Figure 1 needs more explanation. You mention the ultrasound hardware, then refer to the image, which is of an actual muscle thickness image. Either move the image to later in the manuscript where the image evaluation takes place, or include far more detail regarding figure 1 in the same paragraph as it is in currently. I personally recommend moving that present small paragraph to the beginning of the ‘Muscle thickness measurements’ subsections (currently lines 109-117)

Author response: We agree. Figure 1 moved near the section “Muscle thickness measurements” so that it is better understood, as the reviewer suggested. This meant that Figure 1 now becomes Figure 2 and vice versa.

Comment 7. The ’hundred’ exercise needed more explanation. Currently, the authors state, “In the Hundred exercise, individuals were asked to lift their heads and shoulders towards their hips and then performed the ADIM”. Why is it called ‘the hundred’? It is held for 100 seconds, or does the person do 100 reps etc? (CORRECTION: I do see these instructions in the appendix, however, I would now recommend adding them to the body of the text, and removing the appendix..)

Author response: The appendix is now moved into the main paper, as suggested. Description of the Hundred exercise is also clarified, accordingly (Lines 124-166)

Comment 8. Their also needs to be more information regarding WHEN the images are collected. For example, how many seconds after starting the side place is the image taken? How many images are taken? Are they averaged? Do we know the researchers' reliability (this is esp. important given the authors state in the intro that US imaging is dependent on practitioner skill)?

Author response:  US images were taken about during the 4-6th second of the 10s repetition (or near the middle of the repetition duration) when individuals performed the ADIM manoeuvre (for exercises, only). We took three images (corresponding to three reps) and then took the average. The reliability of our experimental set up has been reported in a previous publication and we now provide relevant information (Lines 170-183).

Comment 9. The statistical analysis is good, though I also would like to see post hoc (pairwise) effect sizes

Author response:

In most cases, we did not have pairwise comparisons, because we did not have significant interactions. Partial eta squared values (as an estimate of effect sizes) are provided for the significant main effects, in the ANOVA results.

Results:

Comment 10. The authors can reduce the number of significant digits (esp for F scores) to improve readability.

Author response:

We have reduced the number of digits in the F scores, as requested.

Comment 11. Figure 3 can be greatly improved by providing brackets (or another illustration) to highlight significant differences (with ESs) between exercises. This would make the figure an excellent way of summarizing the entire results section!

Author response:

We added asterisks to illustrate group differences and another symbol to indicate exercise differences.

Round 2

Reviewer 2 Report

The authors have done a good job of addressing my comments/concerns. 

I have no further issues and endorse publication.